# Leonurine Reduces Oxidative Stress and Provides Neuroprotection against Ischemic Injury via Modulating Oxidative and NO/NOS Pathway

**DOI:** 10.3390/ijms231710188

**Published:** 2022-09-05

**Authors:** Ziteng Deng, Jiao Li, Xiaoquan Tang, Dan Li, Yazhou Wang, Shengxi Wu, Kai Fan, Yunfei Ma

**Affiliations:** 1College of Veterinary Medicine, China Agricultural University, Beijing 100193, China; 2Department of Neurobiology, School of Basic Medicine, The Fourth Military Medical University, Xi’an 710032, China

**Keywords:** ischemic stroke, oxidative stress, apoptosis, NO/NOS, leonurine

## Abstract

Leonurine (Leo) has been found to have neuroprotective effects against cerebral ischemic injury. However, the exact molecular mechanism underlying its neuroprotective ability remains unclear. The aim of the present study was to investigate whether Leo could provide protection through the nitric oxide (NO)/nitric oxide synthase (NOS) pathway. We firstly explored the effects of NO/NOS signaling on oxidative stress and apoptosis in in vivo and in vitro models of cerebral ischemia. Further, we evaluated the protective effects of Leo against oxygen and glucose deprivation (OGD)-induced oxidative stress and apoptosis in PC12 cells. We found that the rats showed anxiety-like behavior, and the morphology and number of neurons were changed in a model of photochemically induced cerebral ischemia. Both in vivo and in vitro results show that the activity of superoxide dismutase (SOD) and glutathione (GSH) contents were decreased after ischemia, and reactive oxygen species (ROS) and malondialdehyde (MDA) levels were increased, indicating that cerebral ischemia induced oxidative stress and neuronal damage. Moreover, the contents of NO, total NOS, constitutive NOS (cNOS) and inducible NOS (iNOS) were increased after ischemia in rat and PC12 cells. Treatment with L-nitroarginine methyl ester (L-NAME), a nonselective NOS inhibitor, could reverse the change in NO/NOS expression and abolish these detrimental effects of ischemia. Leo treatment decreased ROS and MDA levels and increased the activity of SOD and GSH contents in PC12 cells exposed to OGD. Furthermore, Leo reduced NO/NOS production and cell apoptosis, decreased Bax expression and increased Bcl-2 levels in OGD-treated PC12 cells. All the data suggest that Leo protected against oxidative stress and neuronal apoptosis in cerebral ischemia by inhibiting the NO/NOS system. Our findings indicate that Leo could be a potential agent for the intervention of ischemic stroke and highlighted the NO/NOS-mediated oxidative stress signaling.

## 1. Introduction

Stroke represents one of the main causes of disability and death worldwide. The pathological subtypes of stroke comprise ischemic stroke and hemorrhagic stroke, in particular, 87% of which is ischemic stroke [1,2]. Oxidative stress and apoptosis are considered to be the key pathogenic mechanisms of the brain injury caused by ischemic stroke [3,4,5]. Though oxidative metabolism is highly essential for the survival of neurons, the consequence of oxidative stress is oxidative damage that occurs when there is an excess of oxygen free radicals (oxidants) or depletion of antioxidants, which can potentially lead to cellular dysfunction, apoptosis, necrosis and/or cell death [3,6]. In physiological conditions, excessive reactive oxygen species (ROS) could be scavenged by endogenous antioxidant defense systems, such as catalase (CAT), superoxide dismutase (SOD) and the glutathione system (GSH), to maintain homeostasis [7,8,9]. Thus, antioxidants and antioxidant defense systems play a critical role in exerting a protective effect on tissues and cells from oxidative damage.

Nitric oxide (NO), a free radical, is produced by nitric oxide synthases (NOS), which converts L-arginine to L-citrulline with the consequent release of NO. Three isoforms of NOS including neuronal NOS (nNOS), endothelial NOS (eNOS) and inducible NOS (iNOS) were identified. Both nNOS and eNOS are considered constitutively expressed (also known as cNOS), modulated by Ca^2+^ activated calmodulin, whereas iNOS is calcium-independent [10]. NO rapidly reacts with the superoxide anion radical [O_2_^−^] to yield the peroxynitrite anion [ONOO^−^], an important component of ROS, which can induce not only oxidative stress but also nitrative stress. Normal production of ONOO^−^ is involved in growth, proliferation and differentiation, whereas abnormal production may cause lipid peroxidation, protein oxidation and DNA damage, leading to cell death via either apoptosis or necrosis depending on the nature and extent of oxidative or nitrative stress [8]. Recently, many studies have evidenced the implication of NO/NOS in a variety of pathophysiological symptoms, including cerebral ischemic stroke, cardiovascular disease and neurodegenerative disorders, such as Alzheimer’s disease, Parkinson’s disease and hypertension [4,10,11]. Therefore, the regulatory balance in the production of NO and its derived species depends not only on the activity of the NOS isoforms but also on the oxidative state of the cell, with both determining the cell fate.

According to the aforementioned, inhibiting the excessive production of ROS and/or regulating the NO/NOS signaling pathway would be a suitable strategy to decrease oxidative stress and apoptosis induced by ischemic stroke. However, few therapeutic drugs and strategies show an effective potential for ischemic stroke to date. Thus, it is necessary to investigate the underlying mechanisms of ischemic stroke and develop new therapeutic drugs to ameliorate the consequences of cerebral ischemia injury.

Herbal medicine has been reported as a promising alternative choice for treating ischemic cerebral injury and thereby, greater attention should be given to natural compounds with wide therapeutic windows, clear pharmacological targets and fewer side effects [12]. Leonurine (C_14_H_21_N_3_O_5_, Leo), one of the active alkaloid compounds purified from Herba Leonuri, is a traditional Chinese herbal medicine with a great deal of biochemical activities including antioxidant, anti-inflammatory and antiapoptotic properties [13,14,15,16]. However, whether Leo plays a neuroprotective role through the NO/NOS pathway and the exact mechanism regarding how the NO/NOS signaling pathway exerts its impact in ischemic stroke are still to be elucidated.

In the present study, given the key role of the NO/NOS signaling pathway in cerebral ischemia events, we hypothesize that Leo, a substance capable of reducing oxidative stress, can exert its neuroprotective effect after cerebral ischemia, at least in part, by modulating the NO/NOS system response. We established an in vivo model of cerebral ischemia stimulated by photochemistry and an oxygen–glucose deprivation (OGD)-induced PC12 cells model in vitro, combined with the pretreatment of L-NAME, an inhibitor of NOS, to explore the effects of NO/NOS signaling on oxidative stress and apoptosis in vivo and in vitro. We then investigated the protective effects of Leo against OGD-induced oxidative stress in PC12 cells and evaluated the molecular mechanism underlying its neuroprotection related to the NO/NOS pathway.

## 2. Results

### 2.1. Behavioral Performance of the Rats after Ischemia and L-NAME Treatment

The behavioral performance of rats was observed at 24 h after the ischemic induction. Compared with the control group, there was no obvious change in Sham (*p* > 0.05). Cerebral ischemia resulted in the deficiency of entertainment, higher vigilance, vertical fur and dry hair in rats, but these were significantly reversed by pretreatment with L-NAME (Figure 1A). We further used OFT to measure the anxiety behavior of the rats (Figure 1B,C). When compared with the control group, the duration of the Is significantly decreased by 3.75% (*p* < 0.01) in the inner zone and increased by 49.04% (*p* < 0.01) in the outer zone, toward the corner or along the wall. In addition, the duration of staying in the inner zone was significantly elevated by 48.76% (*p* < 0.05) in the L-NAME compared with the Is.

### 2.2. SOD, MDA, GSH and ROS Levels after Ischemia and L-NAME Treatment

There was no remarkable dissimilarity in the level of SOD, MDA and GSH between the control group and the Sham (*p* > 0.05) both in serum and in brain tissue. Cerebral ischemia caused a significant reduction in SOD activity and GSH content (both *p* < 0.05) and a remarkable dilation of MDA levels (*p* < 0.01 in serum and *p* < 0.05 in brain tissue), compared with the control group (Figure 2A–C). However, the detrimental effects in Is were significantly reversed by pretreatment with L-NAME, a nonselective NOS inhibitor, including enhancement of SOD activity (*p* < 0.01 in serum and *p* < 0.05 in brain tissue) and inhibition of MDA generation (*p* < 0.05 in serum) to a level similar to that in controls. However, there was no significant difference in the content of GSH (*p* > 0.05) between the Is and the L-NAME. Using a ROS detection kit, we determined the expression of ROS in the brain tissue, and the results show that the generation of ROS in the Is increased significantly (*p* < 0.01), compared with the control group; it was nevertheless dramatically restrained by preadministration with L-NAME (*p* < 0.05) (Figure 2D). These data indicate that the oxidative and redox levels were changed by ischemia.

### 2.3. Change in NO/NOS Expression in Serum and Brain Tissue after Cerebral Ischemia

The above results suggest that cerebral ischemia induced oxidative stress and that it was necessary to further investigate the role of the NO/NOS signaling pathway. To explore the changing trend of the NO/NOS signal pathway, the expression of NO/NOS/cNOS/iNOS in brain tissue was detected at 2 h, 6 h, 12 h and 24 h after ischemia. Compared with the control group, there was no significant alteration in the expression of NO/NOS/cNOS/iNOS in the Sham (*p* > 0.05). In the Is, the expression of NO and NOS/cNOS decreased at the first 2 h, and then there was an extremely marked increase at 24 h (*p* < 0.01), whereas iNOS continuously increased at 24 h (*p* < 0.01), compared with the controls (Figure 3B). These results suggest that cerebral ischemia can saliently enhance brain NO content and NOS/cNOS/iNOS protein expression.

Based on the above results, we further evaluated the expression of NO/NOS/cNOS/iNOS in serum and brain tissue at 24 h after ischemic injury. As shown in the histogram (Figure 3A,C), no significant differences were observed in the level of NO (*p* > 0.05) and NOS/cNOS/iNOS (*p* > 0.05) in serum and brain tissue between the control group and Sham. Cerebral ischemia caused an extremely significant amplification of NO generation (*p* < 0.01) and NOS/cNOS/iNOS activity (*p* < 0.01) both in serum and in brain tissue when compared with the controls. In addition, L-NAME pretreatment markedly obviated the increase in NOS, iNOS, cNOS and NO in the serum and brain tissue at 24 h after the cerebral ischemia of rats, even similar to the control group, compared with the Is (*p* < 0.05 or *p* < 0.01). Interestingly, the data above also show that the levels of NO and NOS/iNOS/cNOS in the serum were even higher than that in the brain tissue at 24 h after cerebral ischemia, indicating that NO and NOS/cNOS/iNOS may exert its function in serum.

### 2.4. L-NAME Pretreatment Attenuates the Neuronal Damage Induced by Cerebral Ischemia

As shown in the previous description, ischemic brain injury was located in the frontal lobe, including the motor area and premotor area. The cerebral cortex was divided horizontally into six layers: the molecular, external granular, external pyramidal, internal granular, internal pyramidal and multiform layers [17]. Nissl staining results show, in the Sham, that dark blue stained granular Nissl bodies were mainly distributed around the cytoplasm of cortical neurons, with no obvious abnormality in histomorphology and were regularly arranged. In the Is, the neurons in the molecular, external granular and external pyramidal layers were damaged to a different degree. A large number of Nissl bodies were lost, showing that the cortical neurons in the center of infarction were liquefied and necrotic, as well as that the morphological structure was abnormal, and the arrangement of cells was disorderly and loose. In contrast with the Is, pretreatment with L-NAME markedly attenuated the neuronal damage induced by cerebral ischemia, including the reduction in the number of atrophic or necrotic neurons, the partial improvement of morphological structure integrity and more regular cell arrangement (Figure 4A).

To further estimate the degree of neuron damage and the effect of L-NAME on the density of neurons induced by ischemia, NeuN-positive neurons in the cerebral cortex of rats were stained by immunohistochemistry (Figure 4B,C). Compared with the control group, the density of NeuN-positive neurons in the Sham almost did not change (*p* > 0.05), while that of the Is significantly decreased by 30.56% (*p* < 0.01). Interestingly, preadministration with L-NAME prior to ischemia, the density of NeuN-positive neurons extremely increased by 34.57% (*p* < 0.01) when compared with the Is. The results above suggest that L-NAME can protect neurons from damage following cerebral ischemia through ameliorating the necrosis, morphological structure as well as cell density.

### 2.5. L-NAME Pretreatment Affects Bax and Bcl-2 Protein Expression after Cerebral Ischemia

For the validation of neuronal apoptosis, the expression of Bax and Bcl-2 in different treatment groups was examined at 24 h after cerebral ischemia by Western blot analysis. No significant difference in Bax and Bcl-2 (both *p* > 0.05) expression was found between the Sham and the controls. In the Is, the expression of Bax significantly increased by 42.55% (*p* < 0.01) while the Bcl-2 generation remarkably decreased by 18.18% (*p* < 0.01), and the ratio of Bax/Bcl-2 was obviously higher than the control group (*p* < 0.01). We further assessed whether L-NAME protected neurocytes against the change in Bcl-2 and Bax levels. Relative to the Is group alone, treatment by L-NAME significantly impeded Bax expression by 13.43% (*p* < 0.01), whereas the generation of Bcl-2 was markedly augmented by 9.03% (*p* < 0.05), and the ratio of Bax/Bcl-2 was obviously lower than the Is (*p* < 0.01) (Figure 5A–D).

In summary, the above results indicate that NO/NOS mediates oxidative stress and plays a promoting role in neuronal damage. To further identify the role of the NO/NOS signaling pathway in oxidative stress and investigate the neuroprotective mechanisms of Leo on cerebral ischemia, a model of OGD-induced PC12 cells was subsequently established to simulate cerebral ischemic-like conditions.

### 2.6. Effects of Leonurine on Oxidative Stress Levels in OGD-Induced PC12 Cells

To explore the function of Leo on oxidative stress in the context of OGD-induced injury in PC12 cells, we investigated its effect on the intracellular ROS and protein oxidative damage. Under normal physiology, ROS are regulated by endogenous antioxidant defense mechanisms to maintain homeostasis, and an increased ROS level within the cell reflects multiple intracellular metabolic disorders. Therefore, we examined the effect of Leo on ROS levels. As shown in Figure 6A,B, there was a very low content of ROS in the control group; however, the ROS level in the OGD group increased by 4.45% (*p* < 0.01) and was higher than the normal controls. In contrast, it reduced markedly by 2.98% (*p* < 0.01) and 4.34% (*p* < 0.01) when the PC12 cells were pretreated with LEO-MID (100 μg/mL) and LEO-HIGH (200 μg/mL), respectively. A similar effect was observed in the group of L-NAME, an inhibitor of NOS, which decreased by 3.16% (*p* < 0.01). However, there was nevertheless no significant difference between LEO-LOW (50 μg/mL) and the OGD group, which only reduced by 0.74% (*p* = 0.249).

Oxidative stress induced by OGD is one of the important causes of PC12 cell death [18]. Figure 6C,E shows the effects of Leo on SOD and GSH contents in all groups. Compared with the normal control group, OGD reoxygenation remarkably caused a reduction in SOD and GSH activity by 13.76% and 30.39% (both *p* < 0.01). The Leo pretreatment group at a dose of 100 μg/mL and 200 μg/mL produced a significant enhancement in the levels of SOD by 8.68% (*p* < 0.05) and 13.37% (*p* < 0.01) and GSH by 18.78% and 38.05% (*p* < 0.01), respectively, although the LEO-LOW group showed few obvious differences when compared with the OGD group. Consistent with the Leo group, preadministration with L-NAME significantly increased the contents of SOD and GSH by 11.10% (*p* < 0.01) and 24.39% (*p* < 0.05), respectively.

MDA, as a biomarker for oxidative stress, is a product of lipid peroxidation. As shown in Figure 6D, OGD largely up-regulated the level of MDA by 36.67% (*p* < 0.01) when compared with the normal control group. Middle and high dose Leo significantly prevented the generation of MDA by 19.5% and 23.83% (both *p* < 0.01), respectively, compared with the OGD group. A similar but less marked effect was observed in the LEO-LOW group compared with the OGD group. In addition, a positive group of pretreatment with L-NAME remarkably inhibited the content of MDA, similar to the LEO-MID group, when compared with the OGD group. The results above suggest that Leo showed an antioxidant function to attenuate OGD-induced cell injury.

### 2.7. Effects of Leonurine on the NO/NOS System in OGD-Induced PC12 Cells

NO, as a key gas messenger molecule in oxidative stress, is regulated by the expression and activity of NOS [19]. Herein, we further investigated whether Leo interfered with the generation of NO and NOS/iNOS/cNOS. After incubation under 2 h OGD and cultured in normal conditions at different time points (2 h, 6 h, 12 h and 24 h), the levels of NO and NOS/iNOS/cNOS in PC12 cells were determined. In Figure 7A, we found that OGD resulted in a significant decrease in NO at 2 h by 14.32% (*p* < 0.05) and then increased continuously with culturing time at 24 h (*p* < 0.01). Consistent with NO at the first 2 h, the contents of NOS and cNOS significantly decreased (*p* < 0.05 and *p* < 0.01, respectively) but increased to the peak at 12 h and then declined with culturing time at 24 h. Surprisingly, the expression of iNOS seemed to be different from others, representing a peak early (at 6 h, *p* < 0.01) and gradually declined with culturing time at 24 h; however, its expression was significantly lower than the others at all time points.

According to the above results, different doses of Leo and L-NAME were preadministered prior to OGD injury, and the highest expressions of NO/NOS/cNOS/iNOS (24 h, 12 h, 12 h, 6 h, respectively) were evaluated. As described in Figure 7B–E, Leo intervention can remarkably suppress NO/NOS production. Specifically, following middle and high concentrations of Leo pretreatment, the levels of NO/NOS/cNOS/iNOS were markedly depressed at their peak time points, and the LEO-HIGH group almost returned to normal level, even though no significant difference was found in the LEO-LOW group. Treatment with L-NAME also abolished the increased expression of NO/NOS/cNOS/iNOS at their peak state. These findings suggest that Leo is involved in the signaling pathway of NO/NOS after OGD injury.

### 2.8. Effects of Leonurine on the Apoptosis of OGD-Induced PC12 Cells

Accumulating studies have indicated that apoptosis plays a crucial role in neuronal death in ischemic brain injury [20]. To address whether Leo has direct protective roles in PC12 cells following OGD injury, an AO/EB staining and flow cytometry method was used to detect the apoptosis. The results from the AO/EB staining and flow cytometry in Figure 8A–C show that no obvious apoptosis was observed in the control group. However, early apoptotic cells with yellow-green fluorescence and late apoptotic cells with orange or red fluorescence were detected in the OGD group. Along with this, OGD-induced cells exhibited round, slender morphologies as well as concentrated and localized red in the nuclear, with a significantly higher apoptotic rate (27.11%, *p* < 0.01) than in the normal controls. It seemed that 200 μg/mL (LEO-HIGH) of Leo possessed the strongest antiapoptotic effect versus the others (LEO-HIGH, 50 μg/mL and LEO-MID, 100 μg/mL), by notably attenuating the increased apoptosis of OGD-induced PC12 cells, including more normal cells and few early and late apoptotic cells, with a saliently lower (*p* < 0.01) apoptotic rate, even similar to the controls. Cell characteristics in the LEO-LOW exhibited fewer late apoptotic cells, although no significant difference in the rate of cell apoptosis was observed when compared with the OGD group. Additionally, the number of late apoptotic cells in the LEO-MID was even fewer than that of the LEO-LOW, and the cell apoptotic rate was significantly lower (*p* < 0.01).

Further, Western blot was performed to determine whether Leo could alter the expression of apoptosis-related factors after OGD-stimulated cells. Relative to the control group alone, OGD markedly induced the increase in Bax expression and the reduced production of Bcl-2 in PC12 cells, with a significantly higher (*p* < 0.01) ratio of Bax/Bcl-2 (Figure 9A–D). Interestingly, administration with Leo notably reversed the changed expression of Bax and Bcl-2. Specifically, the expression levels of Bax protein were dramatically decreased, while the Bcl-2 protein was markedly elevated in both the high and middle dose Leo treatment groups accompanied by an obvious lower ratio of Bax/Bcl-2 than in the OGD group, and the LEO-HIGH group (*p* < 0.01) nearly returned to normal. However, there was no noteworthy difference between the OGD and LEO-LOW groups, although the ratio of Bax/Bcl-2 was remarkable higher. Based on the data above, we concluded that Leo could directly protect PC12 cells from OGD-stimulated apoptosis.

## 3. Discussion

Our current research suggests that Leo ameliorated OGD-induced injury, apoptosis and oxidative stress-related factor levels in PC12 cells. The mechanistic study demonstrates that the protective properties of Leo on OGD-stimulated changes may occur through inhibiting the NO/NOS signaling pathways. In addition, the data from L-NAME, a NOS inhibitor, also illustrated the neuroprotective effect of Leo.

Stroke is considered as the third leading cause of death in adults, with a high rate of morbidity, mortality and disability [21]. Increasing evidence shows that cerebral ischemia is a frequent symptom of stroke [22]. It is well known that ischemia can cause an abnormal behavior in humans as well as in animals, including anxious performance and neuronal damage. In the present research, results from the OFT indicate that ischemic brain injury in rats exhibited insufficient interest and spent more time in the outer zone, suggesting that anxiety-like behaviors were induced following cerebral ischemia. Moreover, Nissl staining and immunohistochemical staining results reveal that in the Is group, a great number of Nissl bodies and NeuN-positive neurons were significantly lost, indicating neuronal damage of a high degree after ischemia induction. However, the detrimental effect caused by cerebral ischemia was obviously restored by pretreatment with L-NAME, since it has a potent neuroprotective activity in vivo. Meanwhile, an OGD-stimulated model of PC12 cells was established to simulate ischemic-like conditions in vitro to explore the potential mechanism by which the signaling pathway was involved after ischemia. Interestingly, Leo significantly ameliorated cell viability in a concentration-dependent manner without obvious cytotoxicity in the OGD-stimulated PC12 cells. A similar effect was also observed by preadministration with L-NAME. The results above therefore show that Leo may have a potential neuroprotective potency in ischemic brain injury.

The results from various investigations show that, oxidative stress plays a crucial role in the ischemia/reoxygenation process [23]. It is widely accepted that SOD is frequently regarded as an antioxidative enzyme in antioxidant defense systems against cellular and tissue damage induced by cytotoxic reactive oxygen species and is usually utilized to evaluate the extent of damage in these processes in vivo [24]. It is also reported that SOD shows the function to catalyze the reduction in O^−^ to H_2_O_2_ [25]. MDA is a lipid peroxide product of reactive oxygen species attacking protein, DNA and polyunsaturated fatty acid PUFA, and therefore commonly acts as a biomarker of oxidative stress [26]. Glutathione, an antioxidant, consists of tripeptides of cysteine, glutamic acid and glycine, and the most common form is GSH, which can scavenge free radicals by reducing hydrogen peroxide and lipid peroxide [27]. Thus, SOD, MDA and GSH play a central role in the oxidative stress response. In our observations, the ischemia-induced rats as well as OGD-stimulated PC12 cells exhibited a decline in SOD and GSH expression and an augment in the MDA contents compared with the normal controls. In contrast, preadministration with L-NAME and Leo showed potent functions against ischemia- and/or OGD-induced changed antioxidant enzyme (SOD) activities and maintained the generation of MDA and GSH at normal levels. It is worthy to note that the protective effect at the highest dose of Leo is higher than L-NAME, a well-accepted NO blocker agent. In the literature, it is well reported that ROS, a group of mediators of mitochondria, DNA repair enzymes and transcription factors, are conducive to oxidative damage [28,29,30]. ROS generation reduction is one of the strategies to prevent neurons from damage after an oxidative stress event. As mentioned in our data, oxidative stress caused excessive ROS production both in rat brain tissue and PC12 cells. However, ROS levels significantly down-regulated in a dose-dependent trend by pretreatment with Leo, especially in the LEO-HIGH group as well as in the L-NAME group. According to the data above, we therefore propose that Leo can exert antioxidant properties to improve oxidative stress-induced injury.

To the best of our knowledge, nitric oxide (NO) liberated from L-arginine in a condition catalyzed by NO synthases (NOS) react with O_2_^−^ to generate peroxynitrite (ONOO^−^) [31]. Under oxidative stress situations, the pathway of activation of NOS and generation of NO are involved. Within the heart, the isoform of NOS contains at least two types, including cNOS and iNOS [32]. nNOS and eNOS, dependent on a rise in tissue calcium concentration for activity, are regarded as constitutively expressed proteins (cNOS). A large number of studies from animal and cell models show that NO is mainly generated by nNOS in the early phase of cerebral ischemia to induce brain insult while, in the late stage, NO is predominantly induced by iNOS. A previous study investigated that 7-NI, an inhibitor of nNOS, alleviated the blood–brain barrier disorder after transient focal ischemic brain injury, and it could also ameliorate neuronal damage as well as reduce the region of brain infarction [33]. In the recent experiment, the expressions of NOS/iNOS/cNOS were detected in brain tissue and PC12 cells at different time points (2 h, 6 h, 12 h and 24 h) and in serum at 24 h after oxidative stress. Consistent with previous work [34,35], the NO/NOS/iNOS/cNOS levels under oxidative stress conditions were eventually increased at 24 h in all groups compared with the controls, although the reduced expression in brain tissue and PC12 cells were determined at the first 2 h. Interestingly, in our observations, the expression of NO/NOS/iNOS/cNOS in serum were markedly higher than that in brain tissue at 24 h after ischemia, and thus in the future, we have to explore the potential mechanism accounting for these events. Additionally, another attractive finding obtained was that the NOS/iNOS/cNOS levels in OGD-stimulated PC12 cells eventually reduced (higher than controls) after they reached their peak activity, which is inconsistent with the data of brain tissue in rats following ischemia. Treatment by L-NAME significantly decreased the NO content and NOS/iNOS/cNOS expressions both in ischemia-induced rats and OGD-stimulated PC12 cells. In in vitro experiments, we show Leo preadministration dramatically down-regulated NO/NOS/iNOS/cNOS contents in a dose-dependent manner at their peak time points, especially in the LEO-HIGH group, which nearly returned to the normal control level. The results above indicate that, in line with the in vivo experiments, Leo possibly showed its antioxidant functions via inhibiting the NO/NOS signaling pathway.

Existing studies have shown that oxidative stress and ROS production are related to the initiation of apoptosis, which contains obvious morphological and biochemical characteristics [36]. Although the mechanism of apoptosis during ischemia is still not yet fully clear, recent animal studies and clinical observations have shown that the balance between antiapoptotic protein Bcl-2 and proapoptotic protein Bax are associated with the regulation of apoptosis in the brain following ischemia [37,38]. Previous research has suggested that Leo alleviated OGD-induced PC12 cell death via targeting the Cx36/CaMKII signaling pathway [14]. In our present observations, Leo showed alleviative functions on the reduction in cell survival and the enhancement of apoptosis caused by OGD in PC12 cells. In addition, treatment by Leo restored the disturbed expression of Bax and Bcl-2 proteins, especially in the LEO-HIGH group, which was comparable to the control group. Preadministration with L-NAME in OGD-induced PC12 cells showed similar effects when compared with the LEO-HIGH group. The findings from the present study suggest that Leo may act as a neuroprotective agent by modulating apoptosis after oxidative stress.

There are also some limitations in our present study. Firstly, Leo pretreatment was found to regulate the levels of oxidative stress-related factors by inhibiting NO/NOS expression. However, it was not fully clear whether other molecules and pathways were involved. Secondly, inflammatory factors were also associated with cerebral ischemia. We are yet to demonstrate whether Leo administration simultaneously blocked the NO/NOS pathway and alleviated inflammation such as TLR/TRAF/NF-κB pathways to improved cell apoptosis. This will be explored in future experiments. Thirdly, we only used the OGD-induced PC12 cell model in vitro to determine the antioxidant and antiapoptotic functions of Leo, as verified by L-NAME at the same time. Consequently, further research is necessary to estimate the neuroprotective effect of Leo on in vivo ischemia. Moreover, the therapeutic efficacy and time window of Leo in an ischemic animal model also need to be assessed in future observations.

## 4. Materials and Methods

### 4.1. Animals and Focal Cerebral Ischemia Model

Adult male Sprague Dawley rats (8 weeks old; weighing 200–230 g; Beijing Vital River Laboratory Animal Technology Co., Ltd., Beijing, China) were employed in the present study. The rats were individually housed in cages with a controlled temperature (20 ± 3 °C), 60 ± 5% humidity, 12 h light/dark cycle and ad libitum access to water and food. All experimental protocols were approved by the Committee for the Care and Use of Experimental Animals, China Agricultural University. Every effort was made to minimize the suffering and number of animals used.

The rats were randomly divided into four groups: (1) CON: control; (2) Sham: control treated with Rose Bengal, submitted to surgery without laser illumination; (3) Is: submitted to surgery for cerebral ischemia model; (4) L-NAME: treated with L-NAME, submitted to cerebral ischemia model. Focal cerebral ischemia was induced by a slight modification of photochemical model as per the previous description [39]. Briefly, a light-sensitive dye Rose Bengal (Sigma, Burlington, MA, USA) was freshly prepared by dissolving in saline (20 mg/mL saline) and kept protected from light until use. Rats were slowly injected through a tail vein with Rose Bengal dye (50 mg/kg body weight) prior to being anaesthetized with inhaled isoflurane (ZS Dichuang Science Technology Development Co., Ltd., Beijing, China) and secured in a stereotaxic frame in which isoflurane was continuously delivered via a nose cone. After a small incision was made on the scalp, a craniotomic window (3 mm × 4 mm) was made over the motor cortex with the center at the coordinate of 2 mm posterior to the bregma and 2 mm lateral to the midline. A cold laser beam of 1.5 mm diameter and 560 nm wavelength (Bjtoptime Science Technology Co., Ltd., Beijing, China) was stereotactically positioned at the middle of the craniotomic window and illuminated for 25 min to induce focal cerebral ischemia. The Sham were subjected to the same procedures except for laser illumination. In the L-NAME group, after the rats were intraperitoneally injected with a dose of 1% L-NAME (10 mg/kg body weight, diluted with saline) for 20 min, they were treated for cerebral ischemia, and other groups were intraperitoneally pretreated with saline instead of L-NAME. All rats were able to survive until they were sacrificed in the present study.

### 4.2. Behavioral Test

Neurobehavioral test was determined by using the open-field test (OFT) 24 h after the operation. OFT was conducted as previously described [40]. The rats were transferred to the testing apparatus, which was an illuminated, soundproofed box (100 cm × 100 cm × 50 cm) with black inner walls. The bottom surface of the box was divided into 25 squares (20 cm × 20 cm). The 16 squares adjacent to the walls were defined as outer zone, whereas the other nine squares were defined as inner zone. Using 75% alcohol as disinfectant, the floor surfaces and walls of the apparatus were carefully sterilized and then dried prior to the next animal test. Each rat was placed onto a corner square of the arena and allowed to freely explore the open field for 5 min per trial. The total distance traveled in the OFT and time spent in inner zone and outer zone were quantified by the ANY-maze video tracking system (Stoelting Co., Wood Dale, IL, USA), and general behaviour was evaluated as spending time in inner and outer zone of the testing apparatus.

### 4.3. Cell Culture and OGD Model

The neuron-like rat pheochromocytoma cell line PC12 cell was obtained from Boster Biological Technology Co., Ltd. (Wuhan, China) and cultured in Dulbecco’s modified Eagle medium (DMEM) (Gibco, Waltham, MA, USA) containing 10% fetal bovine serum and 100 U/mL antibiotics (penicillin and streptomycin) at 37 °C in 5% CO_2_ in a humidified incubator. To mimic cerebral ischemic-like conditions, a model of OGD-induced PC12 cells was established as previously described [14]. Briefly, the PC12 cells were incubated in glucose-free DMEM, in a hypoxia chamber (Jinfeng Science and Technology Co., Ltd., Beijing, China) containing 95% N_2_ and 5% CO_2_. After incubation for 2 h, the cells were moved to normal cultured conditions (glucose-free DMEM was replaced with DMEM containing 4500 mg/L of glucose) and cultivated for a short period for the subsequent experiments. According to the different evaluation items, the appropriate culture time for PC12 cells after OGD induction was chosen. The PC12 cells were separated into 6 groups: (1) CON: control group with PC12 cells cultured in normal condition; (2) OGD: submitted to OGD model; (3) L-NAME: treated with L-NAME (1 mmol/L), submitted to OGD model; (4) LEO-LOW: treated with the low dose of Leo (50 μg/mL), submitted to OGD model; (5) LEO-MID: treated with the middle dose of Leo (100 μg/mL), submitted to OGD model; (6) LEO-HIGH: treated with the high dose of Leo (200 μg/mL), submitted to OGD model.

### 4.4. Sample Collection and Treatment

In in vivo experiments, a methodology for collection and processing of blood and brain tissue samples following behavioral tests was used as previously described [41]. Two-, six-, twelve- and twenty-four hours later, six rats from each group were humanely sacrificed under isoflurane anesthesia. The blood samples were collected and centrifuged at 2000× *g* for 10 min at 4 °C, and then the serum samples were stored at −20 °C for measurement of oxidative stress index and NO/NOS production. After euthanizing the rats, tissues of the cerebral cortex were dissected, frozen in liquid nitrogen and stored at −80 °C for analysis of oxidative stress and NO/NOS signaling, as well as for Western blot analysis. The other six rats in each group were also deeply anesthetized with chloral hydrate, perfused transcardially with 200 mL of 5 mmol·L^−1^ sodium phosphate (pH 7.4)-buffered 0.9% (*w*/*v*) saline (PBS), followed by 300 mL of 4% (*w*/*v*) formaldehyde in 0.1 mol·L^−1^ sodium phosphate buffer (pH 7.4). The brains were removed to cut into several blocks and postfixed with the same fixative for one day at 4 °C. After cryoprotection with 30% (*w*/*v*) sucrose in PBS, the blocks were sectioned into 30-µm-thick coronal slices on a freezing microtome, which were placed in PBS for subsequent histochemistry and immunohistochemistry. In in vitro experiments, the method of sample collection was the same as that previously described [14].

### 4.5. Oxidation and Antioxidant Test

The degree of lipid peroxidation in the serum and brain tissue of rats and PC12 cells was determined by detecting the ROS and MDA levels. The antioxidant capacity of the serum, brain tissue and cell was evaluated by analyzing the SOD activity and GSH content. The fluorescence value of ROS production was measured by chemiluminescence method, using ROS detection kit (E004, NJJCBio Inc., Nanjing, China). MDA content was measured by the thiobarbituric acid method, using MDA detection kit (A003-2, NJJCBio Inc.). SOD activity was determined by the xanthine oxidase method, using SOD detection kit (A001-1, NJJCBio Inc.). GSH content was quantified by the colorimetric analysis, using GSH detection kit (A006-1, NJJCBio Inc.). All assays were conducted according to manufacturer instructions.

### 4.6. Measurement of NO and NOS Content

NO production in serum, brain tissue and cell were detected, respectively, by using the nitrate reductase method and a NO assay kit (A012, NJJCBio Inc.), according to manufacturer instructions. The level was expressed as µmol/g protein. The activities of total NOS, cNOS and iNOS were measured using NOS assay kit (A014-1, NJJCBio Inc.) in accordance with manufacturer instructions, and the results were expressed as U/mg protein.

### 4.7. Histochemical and Immunohistochemical Procedure

Nissl staining was conducted to examine the neuronal damage or survival in cerebral cortex. After rinsing with 5 mM PBS, the frozen sections were mounted on gelatin-coated glass slides, air-dried, then dehydrated with a series of 50, 70, 95 and 100% alcohol solutions and stored in 100% alcohol overnight. After defatting by xylene for 24 h, the slides were rehydrated and rinsed in distilled water, subsequently stained using 0.1% cresyl violet solution for 30 min, then differentiated in 95% ethyl alcohol for 2–3 min, dehydrated again in 100% alcohol and finally cleared in xylene.

Immunohistochemical staining was employed to assess the change in morphology of cerebral cortex in rats. Frozen sections were rinsed in PBS and then incubated in 1% hydrogen peroxide solution for 30 min to quench the endogenous peroxidase reactivity. All the following incubations described hereafter were performed at room temperature and followed by a rinse with 5 mM PBS at pH 7.4 containing 0.3% Triton X-100 (PBS-X). The sections were incubated overnight in a humidified chamber with a mouse monoclonal antibody against NeuN (1:500, Millipore, Billerica, MA, USA) in PBS-X containing 0.12% lambda-carrageenan, 0.02% sodium azide and 1% donkey serum (PBS-XCD). After a rinse with PBS-X, the sections were incubated for 2 h with biotinylated donkey anti-mouse IgG (1:100; Jackson, West Grove, PA, USA) in PBS-XCD and then for 1 h with ABC (1:50; Vector, Torrance, CA, USA). After a rinse with PBS-X, the bound peroxidase was finally developed as brown by reaction for 5 min with 0.02% diaminobenzidine-4HCl (DAB) (Sigma, Burlington, MA, USA) and 0.0001% H_2_O_2_ in 50 mM Tris-HCl (pH 7.6). All the stained sections were mounted onto the gelatinized glass slides, dried in an ethanol series, cleared in xylene and finally, coverslipped.

The sections were observed under a microscope (Ni-U, Nikon, Tokyo, Japan), and the immunoreactivity intensities of NeuN were determined using Image-Pro plus 6.0 (Media Cybernetics, Bethesda, MD, USA). The cytoarchitectonic areas of cerebral cortex were determined using the rat brain atlas [42]. Three slices were randomly selected for each group (five regions per slice) to count the neuronal numbers.

### 4.8. Western Blot Analysis

Protein expressions of Bax/Bcl-2 in cerebral cortex and PC12 cells were assessed using Western blot analysis as previously described [14,43]. Briefly, total protein was extracted using a total protein extraction kit (Biochain, Hayward, CA, USA) and quantified using a bicinchoninic acid (BCA) protein assay kit (78510, Pierce, Rockford, IL, USA). Following sodium dodecyl sulfate polyacrylamide gel electrophoresis (SDS-PAGE, 8–12%), equivalent amounts of proteins were transferred onto polyvinyl difluoridine (PVDF) membranes (Millipore, Billerica, MA, USA) at 250 mA for 90 min. After rinsing 3 times with TBST, the polyvinylidene fluoride membranes were placed in a 5% nonfat milk blocking solution on a shaking table at room temperature for 2 h. Subsequently, the membranes were incubated overnight at 4 °C with one of the following primary antibodies: rabbit polyclonal antibody against Bax (1:1000; Sigma, Burlington, MA, USA), rabbit polyclonal antibody against Bcl-2 (1:1000; Sigma, USA) or mouse monoclonal antibody against β-actin (1:1000; 50201; Kemei Borui Science and Technology Co., Ltd., Beijing, China). The membranes were then incubated for 1 h at 37 °C with horseradish peroxidase-conjugated secondary antibodies (HRP-Goat-anti-Rabbit IgG, 1:5000, Kemei Borui Science and Technology Co., Ltd., Beijing, China; HRP-Goat-anti-mouse IgG, 1:10,000, Earth-OX, Millbrae, CA, USA, respectively), and proteins were detected using enhanced chemiluminescence (ECL) substrate (Millipore, Billerica, MA, USA) according to manufacturer instructions. Finally, protein bands were visualized using a chemiluminescence system (5200, Tanon Science & Technology Co., Ltd., Shanghai, China).

### 4.9. AO/EB Staining

Cell apoptosis was detected by double staining with acridine orange (AO) and ethidium bromide (EB). The PC12 cells were cultured on glass coverslips as previously described and then washed with PBS. A mixture of 100 μg/mL AO and 100 μg/mL EB (AO/EB, Sigma, St. Louis, MO, USA) was added to each coverslip, and each coverslip was covered on a slide and kept in a dark place for 15 min, according to manufacturer instructions. The morphology of the apoptotic cells was observed under a fluorescent microscope (Ni-U, Nikon, Tokyo, Japan).

### 4.10. Flow Cytometry Analysis

Cell apoptotic rate was evaluated using an Annexin V-FITC/PI apoptosis detection kit (KeyGen BioTech Co., Ltd., Nanjing, China). After trypsinization, single-cell suspensions were extracted and washed with PBS. The cells were resuspended in 500 µL of binding buffer and stained with Annexin V-FITC and PI for 15 min according to the manufacturer instructions. The samples were then analyzed using a flow cytometer (BD, Franklin Lakes, NJ, USA) with a maximal excitation wavelength at 488 nm and emission at 530 nm.

### 4.11. Statistics

Statistically, no less than six rats in each group were required for statistical significance, and each in vitro experiment was repeated three times. The data were evaluated using one-way ANOVA, followed by Tukey’s post hoc test with SPSS 21.0 (IBM Inc., Chicago, IL, USA). Before analysis, the normal distribution of all data and homogeneity of variances were verified. All the data were expressed as mean ± standard error mean (SEM). A *p* value < 0.05 was considered statistically significant.

## 5. Conclusions

Taken all together, the evidence from different data cumulatively indicate that L-NAME and Leo can protect neurons from ischemic injury. This property of Leo may be associated with reducing oxidative stress and ameliorating neuronal cell apoptosis through blocking the NO/NOS signaling pathway. Therefore, we conclude that Leo may be conducive to the development of a promising agent against cerebral ischemia.

## Figures and Tables

**Figure 1 ijms-23-10188-f001:**
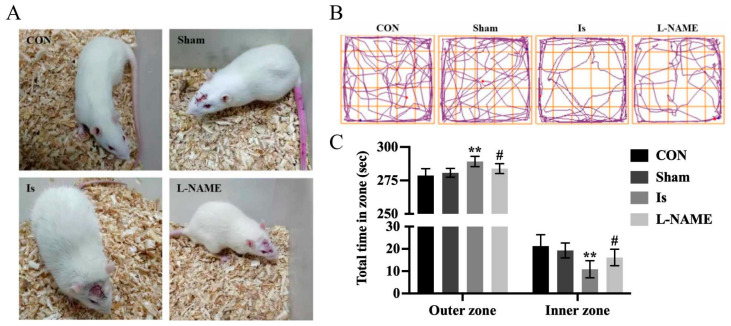
Behaviour of rats after cerebral ischemia and L-NAME treatment according to the OFT. (**A**) Rat behaviour and physical appearance in each group after ischemic induction. (**B**) The trajectory of rats in the OFT. (**C**) The duration of staying in the inner and outer zone in the OFT. Data are presented as mean ± SEM (*n* = 6 per group). ** indicates that the difference between the Is and the CON is extremely significant (*p* < 0.01); # indicates that the difference between the L-NAME and the Is is significant (*p* < 0.05). CON: control group; Sham: sham-operated control group; Is: ischemic brain stroke group; L-NAME: pretreatment L-nitroarginine methyl ester; (L-NAME) + ischemic brain stroke group; the same below in vivo.

**Figure 2 ijms-23-10188-f002:**
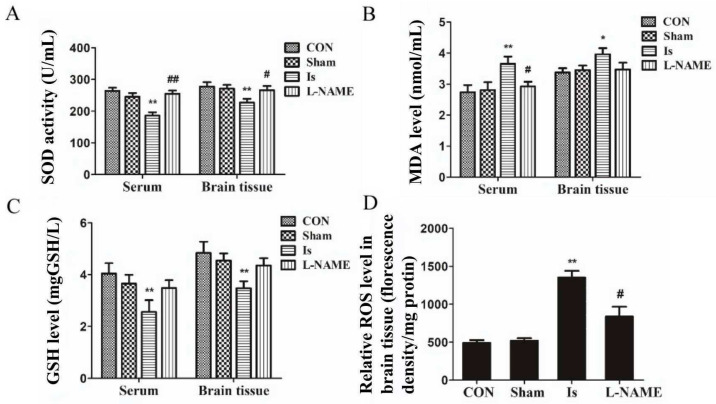
Oxidative stress levels in serum and brain tissue of rats at 24 h after ischemic induction. Effect of ischemic induction on the content of SOD (**A**), MDA (**B**) and GSH (**C**) in serum and brain tissue, and the content of ROS in brain tissue (**D**). Data are presented as mean ± SEM (*n* = 6 per group). * or ** indicates that the difference between the Is and the CON is significant (*p* < 0.05) or extremely significant (*p* < 0.01); # or ## indicates that the difference between the L-NAME and the Is is significant (*p* < 0.05) or extremely significant (*p* < 0.01).

**Figure 3 ijms-23-10188-f003:**
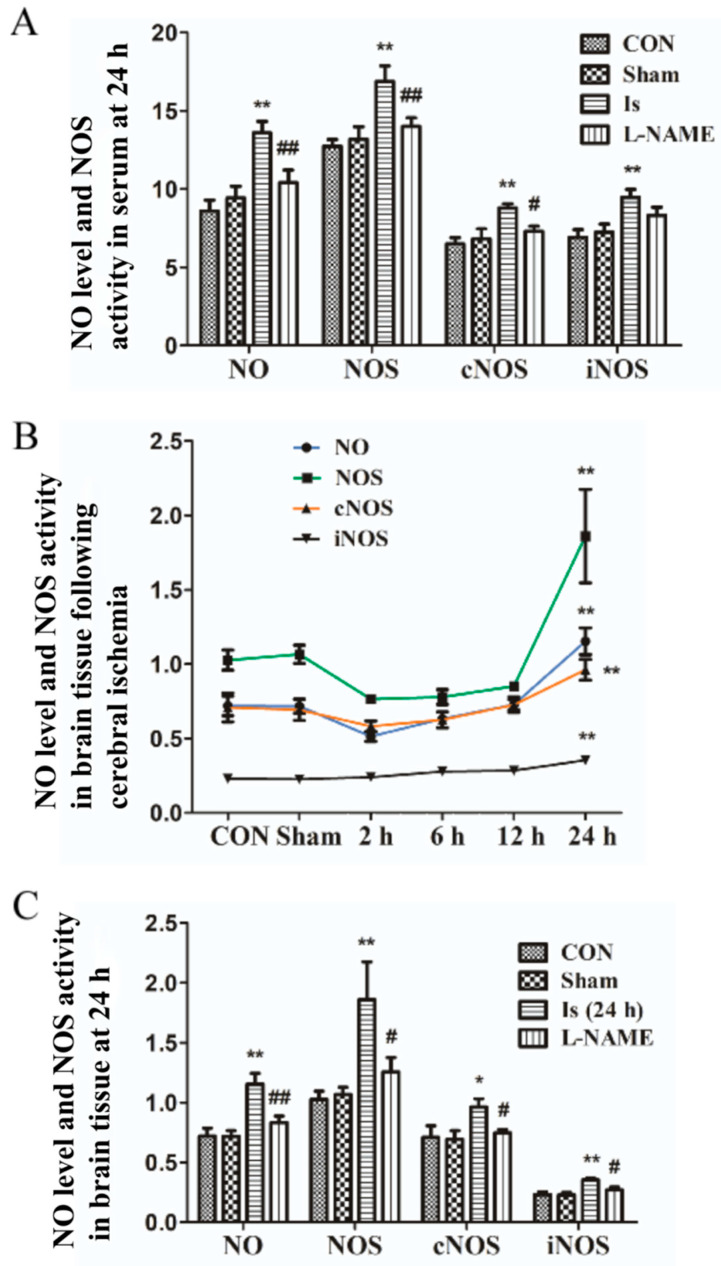
Expression of NO/NOS in serum and brain tissue. (**A**) The content of NO and NOS/cNOS/iNOS in serum at 24 h after ischemic induction in each group. (**B**) The content of NO and NOS/cNOS/iNOS in brain tissue at 2 h, 6 h, 12 h and 24 h after ischemic induction. (**C**) The effects of L-NAME on the highest expression of NO and NOS/cNOS/iNOS (at 24 h after ischemic induction) in brain tissue. Data are presented as mean ± SEM (*n* = 6 per group). * or ** indicates that the difference between the Is and the CON is significant (*p* < 0.05) or extremely significant (*p* < 0.01); # or ## indicates that the difference between the L-NAME and the Is is significant (*p* < 0.05) or extremely significant (*p* < 0.01).

**Figure 4 ijms-23-10188-f004:**
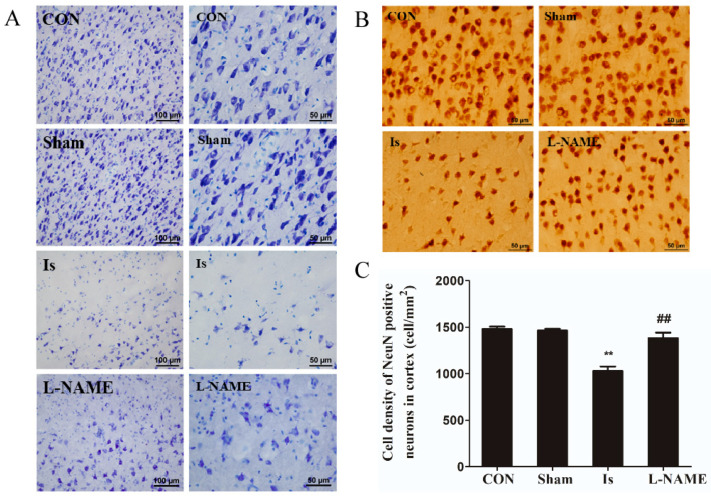
Effect of cerebral ischemia and L-NAME on cortical neurons of rats (**A**) Cortical morphology as depicted by Nissl staining, 20× on the left and 40× on the right. (**B**) Distribution of NeuN-positive neurons in the cerebral cortices by immunohistochemistry staining. (**C**) Cell density of NeuN-positive neurons in the cerebral cortices. Data are presented as mean ± SEM (*n* = 6 per group). ** indicates that the difference between the Is and the CON is extremely significant (*p* < 0.01); ## indicates that the difference between the L-NAME and the Is is extremely significant (*p* < 0.01).

**Figure 5 ijms-23-10188-f005:**
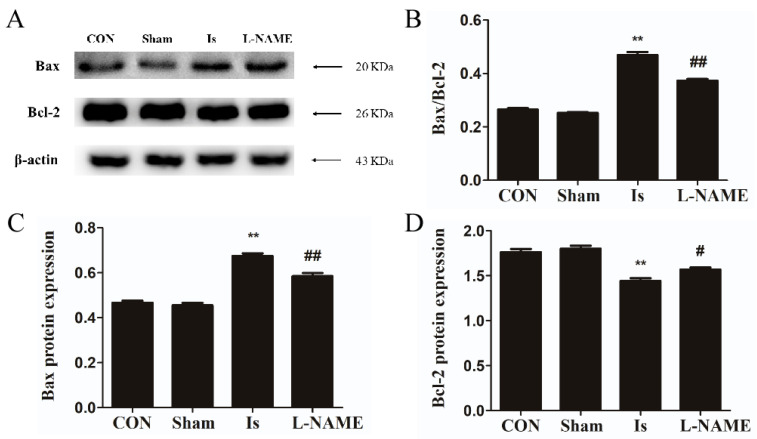
Protein expression of Bax and Bcl-2 at 24 h after cerebral ischemia. (**A**) Protein expression of Bax and Bcl-2 in each group by Western blot analysis. (**B**) The ratio of Bax/Bcl-2 in each group. Image J analysis of Bax (**C**) and Bcl-2 (**D**) levels. The experiments were performed in triplicate. Data are presented as mean ± SEM (*n* = 6 per group). ** indicates that the difference between the Is and the CON is extremely significant (*p* < 0.01); # or ## indicates that the difference between the L-NAME and the Is is significant (*p* < 0.05) or extremely significant (*p* < 0.01).

**Figure 6 ijms-23-10188-f006:**
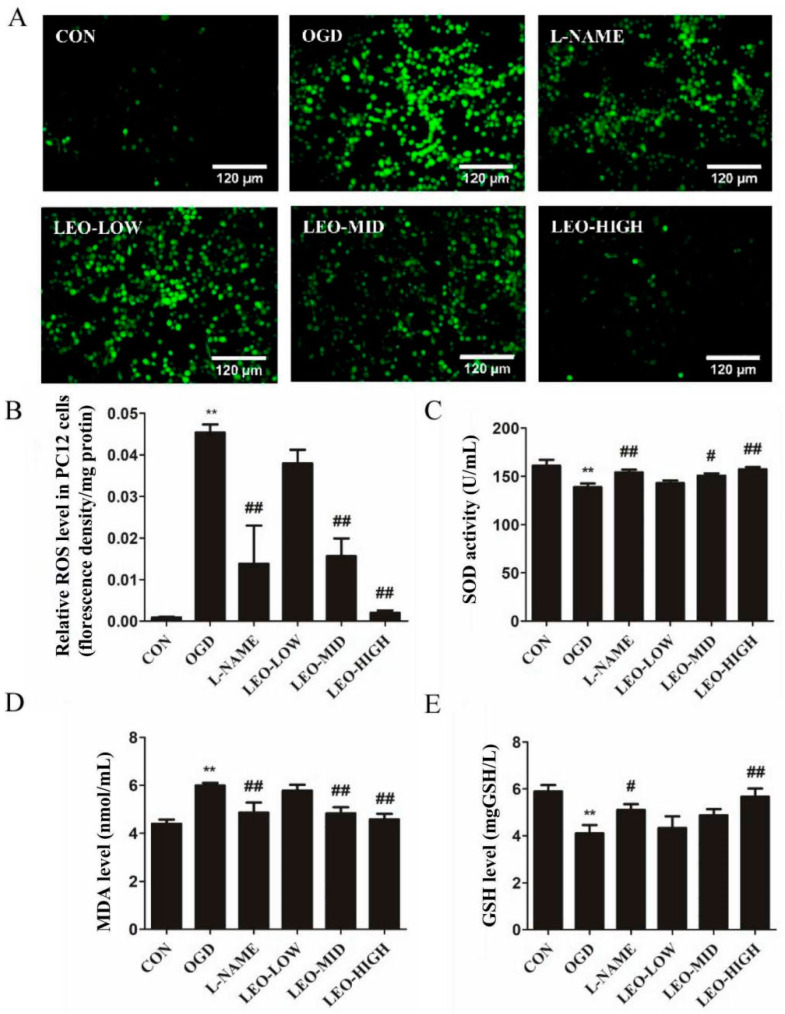
Effects of leonurine on oxidative stress factors in OGD-induced PC12 cells. The fluorescence value of ROS (**A**,**B**) and the contents of SOD (**C**), MDA (**D**) and GSH (**E**) in each group after OGD for 2 h. Data are presented as mean ± SEM. ** indicates that the difference between the OGD and the CON is extremely significant (*p* < 0.01); # or ## indicates significant (*p* < 0.05) or extremely significant (*p* < 0.01) difference compared with the OGD. CON: control group; OGD: OGD model group; L-NAME: the L-NAME group (1 mmol/L); LEO-LOW: low dose of leonurine (50 μg/mL); LEO-MID: middle dose of leonurine (100 μg/mL); and LEO-HIGH: high dose of leonurine (200 μg/mL). The same below in vitro.

**Figure 7 ijms-23-10188-f007:**
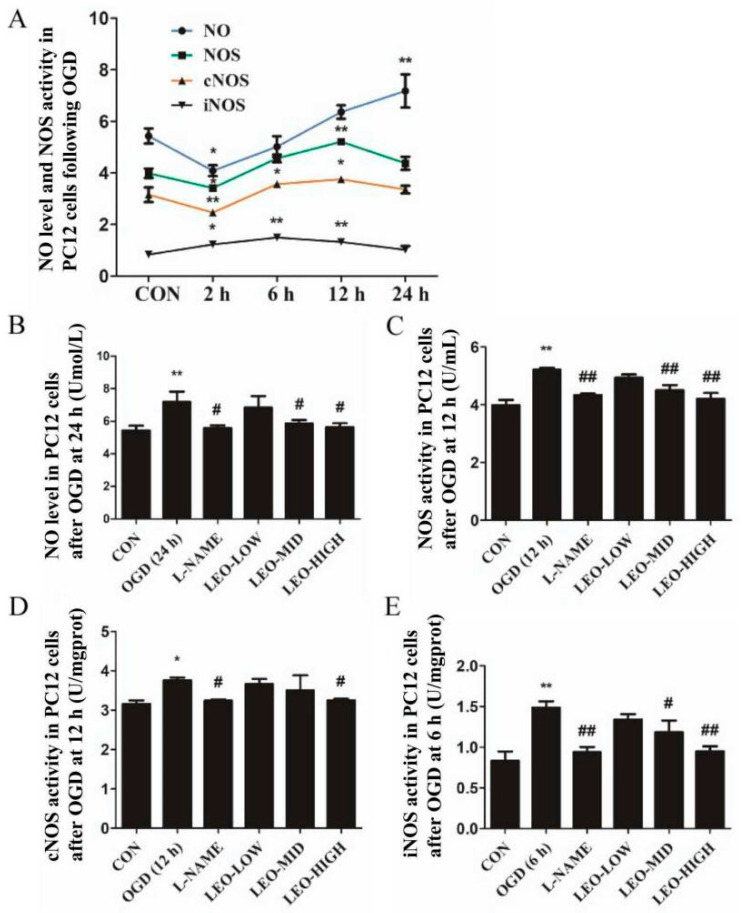
Effects of leonurine on NO and NOS expression levels in OGD-induced PC12 cells. The expressions of NO and NOS/cNOS/iNOS at different times after OGD (**A**). Effects of L-NAME and different doses of leonurine on the highest expression of NO at 24 h (**B**) and NOS at 12 h (**C**)/cNOS at 12 h (**D**)/iNOS at 6 h (**E**). Data are presented as mean ± SEM. * or ** indicates that the difference between the OGD and the CON is significant (*p* < 0.05) or extremely significant (*p* < 0.01); # or ## indicates that significant (*p* < 0.05) or extremely significant (*p* < 0.01) difference compared with the OGD.

**Figure 8 ijms-23-10188-f008:**
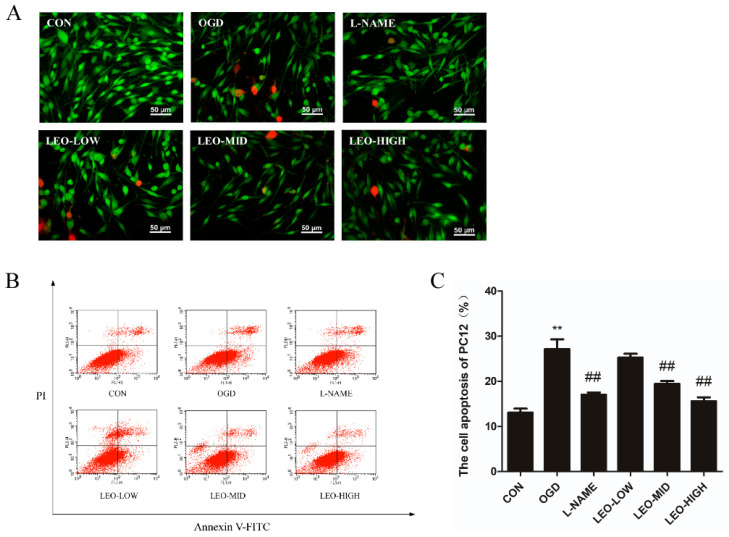
Effects of leonurine on the apoptosis of OGD-induced PC12 cells. (**A**) The apoptosis of OGD-induced PC12 cells by AO/EB staining. (**B**) OGD-induced PC12 cells at different stages of apoptosis of each group by Annexin V/PI and the percentage of apoptotic cells in each group by flow cytometry. (**C**) Data are presented as mean ± SEM. ** indicates that the difference between the OGD and the CON is extremely significant (*p* < 0.01); ## indicates that extremely significant (*p* < 0.01) difference compared with the OGD.

**Figure 9 ijms-23-10188-f009:**
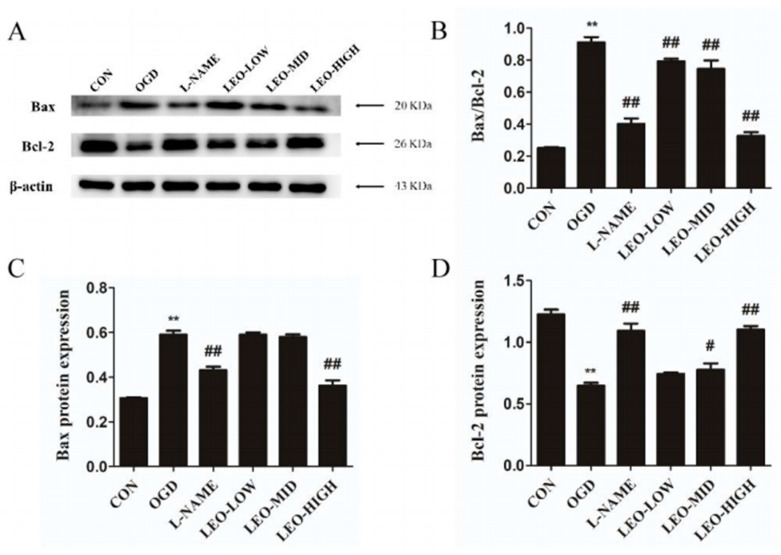
Effects of leonurine on levels of Bax and Bcl-2 after OGD. (**A**) Protein expression of Bax and Bcl-2 in each group at 24 h after OGD by Western blot analysis. (**B**) The ratio of Bax/Bcl-2 in each group. Image J analysis of Bax (**C**) and Bcl-2 (**D**) levels. The experiments were performed in triplicate. Data are presented as mean ± SEM. ** indicates that the difference between the OGD and the CON is extremely significant (*p* < 0.01); # or ## indicates that significant (*p* < 0.05) or extremely significant (*p* < 0.01) difference compared with the OGD.

## Data Availability

All figures and data used to support this study are included within this article.

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
