# Peer review of "Leonurine Reduces Oxidative Stress and Provides Neuroprotection against Ischemic Injury via Modulating Oxidative and NO/NOS Pathway"

_ijms, 2022, doi:10.3390/ijms231710188_

Round 1

Reviewer 1 Report

In this study, the authors aim to investigate whether Leo could provide protection through nitric oxide (NO)/nitric oxide synthase (NOS) pathway. In the first series of experiments, the authors explored the effects of NO/NOS signaling using L-NAME on oxidative stress and apoptosis in in vivo and in vitro models of cerebral ischemia. Subsequently, the protective effects of Leo and L-NAME were evaluated against oxygen and glucose deprivation (OGD)-induced oxidative stress and apoptosis in PC12 cells. Overall, the data suggest that L-NAME and Leo can protect neuron from ischemic injury. This property of Leo may be associated with reducing oxidative stress and ameliorating neuronal cell apoptosis through blocking NO/NOS signaling pathway.

Major comments:

1. NO is commonly regarded as a protective molecule in most conditions. However, in this case, it is responsible causing oxidative stress, probably through formation of peroxynitrite. Hence, direct measurement of peroxynitrite levels would be highly valuable to valid this hypothesis.

2. NOS is known to produce NO, however under pathological conditions, NOS can be uncoupled and produce superoxide. Additional experiments to determine NOS uncoupling and possibly NOS recoupling by LEO treatment will increase the novelty of the study

3. LEO is a well-known antioxidant, does it scavenge free radical directly? Would this have any additional beneficial effects?

4. Inflammation is also an important contributor to the pathogenesis for stroke. Does LEO has any potential anti-inflammatory effects?

5. Based on the experimental design and data, it appears that LEO and L-NAME has similar mechanistic effects. Have the authors looked at the effects of LEO+L-NAME in the OGD model? Perhaps there could be any NOS-independent effects of LEO?

Author Response

Reviewer #1

Comments and Suggestions for Authors: In this study, the authors aim to investigate whether Leo could provide protection through nitric oxide (NO)/nitric oxide synthase (NOS) pathway. In the first series of experiments, the authors explored the effects of NO/NOS signaling using L-NAME on oxidative stress and apoptosis in in vivo and in vitro models of cerebral ischemia. Subsequently, the protective effects of Leo and L-NAME were evaluated against oxygen and glucose deprivation (OGD)-induced oxidative stress and apoptosis in PC12 cells. Overall, the data suggest that L-NAME and Leo can protect neuron from ischemic injury. This property of Leo may be associated with reducing oxidative stress and ameliorating neuronal cell apoptosis through blocking NO/NOS signaling pathway.

Response: Thank you for your support.

Major comments:

  1. NO is commonly regarded as a protective molecule in most conditions. However, in this case, it is responsible causing oxidative stress, probably through formation of peroxynitrite. Hence, direct measurement of peroxynitrite levels would be highly valuable to valid this hypothesis.

Response: Thanks to you for the comment. It is well-known that NO is produced by NOS, which consists of iNOS, cNOS and nNOS. Our current study mainly focused on NO/NOS levels in serum and brain tissues following cerebral ischemia, because a large number of literature showed that both NO/NOS and oxidative stress were participating in cerebral ischemia, but the relation of them still to be elucidated. It was also reported that appropriate conversion between nitrite and NO could protect brain tissues from ischemic damage. We quite agree to your suggestion that peroxynitrite levels should be directly measured, and it will be tested in the further research.   

  1. NOS is known to produce NO, however under pathological conditions, NOS can be uncoupled and produce superoxide. Additional experiments to determine NOS uncoupling and possibly NOS recoupling by LEO treatment will increase the novelty of the study.

Response: Thank you very much for your question. We have noticed our limitations. There is very little research that LEO treatment could preform an effect on NOS uncoupling or recoupling, and by all appearances, it is a valuable research issue.

  1. LEO is a well-known antioxidant, does it scavenge free radical directly? Would this have any additional beneficial effects?

Response: Thank you for bringing up this issue. Accumulating evidence suggests that LEO exhibits various bioactivities such as antioxidant, anti-apoptotic effects, free radical scavenging and anti-inflammatory effects [1]. Previous study had demonstrated that LEO treatment resulted in reduction of ROS production [2]. Consistent with this finding, our data show that ROS level significantly down-regulated in a dose-dependent trend by pretreatment with Leo, especially in the LEO-HIGH group as well as in L-NAME group, and then attenuated OGD-induced damage in PC12 cells. Our group had also reported that LEO had a protective effect on OGD-induced PC12 cells through targeting the Cx36/CaMKII pathway [3].

[1] Zhu, Y.Z.; Wu, W.J.; Zhu, Q.; Liu, X.H. Discovery of Leonuri and therapeutical applications: From bench to bedside. Pharmacol Ther 2018, 188, 26-35, doi:10.1016/j.pharmthera.2018.01.006.

[2] Loh, K.P.; Qi, J.; Tan, B.K.H.; Liu, X.H.; Wei, B.G.; Zhu, Y.Z. Leonurine protects middle cerebral artery occluded rats through antioxidant effect and regulation of mitochondrial function. Stroke 2010, 41, 2661-2668, doi:10.1161/STROKEAHA.110.589895.

[3] Li, J.; Zhang, Shuang.; Liu, X.X.; Han, D.P.; Xu, J.Q.; Ma, Y.F. Neuroprotective effects of leonurine against oxygen-glucose deprivation by targeting Cx36/CaMKII in PC12 cells. PLoS One 2018, 13, e0200705, doi:10.1371/journal.pone.0200705.

  1. Inflammation is also an important contributor to the pathogenesis for stroke. Does LEO has any potential anti-inflammatory effects?

Response: Thanks to you for the comment. In the present research, it was our limitations that the anti-inflammatory effects of LEO in OGD-induced PC12 cells were not concerned. But however, emerging evidence indicated that LEO produced neuroprotective effects in ischemic stroke, partly through inhibiting inflammatory factor levels [1].   

[1] Jia, M.M.; Li, C.X.; Zheng, Y.; Ding, X.J.; Chen, M.; Ding, J.H; Du, R.H.; Lu, M.; Hu, G. Leonurine Exerts Antidepressant-Like Effects in the Chronic Mild Stress-Induced Depression Model in Mice by Inhibiting Neuroinflammation. Int J Neuropsychopharmacol 2017, 20, 886-895, doi:10.1093/ijnp/pyx062.

  1. Based on the experimental design and data, it appears that LEO and L-NAME has similar mechanistic effects. Have the authors looked at the effects of LEO+L-NAME in the OGD model? Perhaps there could be any NOS-independent effects of LEO?

Response: Thank you very much for your question. Our data show that LEO protects against oxidative stress and neuronal apoptosis in cerebral ischemia by inhibiting NO/NOS system. A similar effect was observed in the group of L-NAME, an inhibitor of NOS. Maybe combined use of them show better effects than single. Additionally, our group had revealed that LEO had neuroprotective effects on OGD-induced PC12 cells also through downregulating the protein expression of Cx36 and pCaMKII/CaMKII [1].

[1] Li, J.; Zhang, Shuang.; Liu, X.X.; Han, D.P.; Xu, J.Q.; Ma, Y.F. Neuroprotective effects of leonurine against oxygen-glucose deprivation by targeting Cx36/CaMKII in PC12 cells. PLoS One 2018, 13, e0200705, doi:10.1371/journal.pone.0200705.

Reviewer 2 Report

The authors would like to show therapeutic effects of Leo for ischemia model both in vitro and in vivo. This manuscript is originally well-written and over all study design is reasonable. There are a few issues to be addressed to strengthen the paper.

1. Study limitation: The protocol (time schedule) to simulate the clinical settings after ischemic insult could be considered. If not, the authors could add the limitation in terms of study design or therapeutic conditions.

2. How about the ischemic penumbra sample, ischemic core sample, and sample far from the ischemia, but not 'brain sample' used in this study. The meaning of the obtained region is critically important.

3. The number of rats used in this study should be described for each evaluation.

Author Response

Reviewer #2

Comments and Suggestions for Authors: The authors would like to show therapeutic effects of Leo for ischemia model both in vitro and in vivo. This manuscript is originally well-written and over all study design is reasonable. There are a few issues to be addressed to strengthen the paper.

Response: Thank you for your support.

Major comments:

  1. Study limitation: The protocol (time schedule) to simulate the clinical settings after ischemic insult could be considered. If not, the authors could add the limitation in terms of study design or therapeutic conditions.

Response: Thank you for your helpful advice. We apologize for the inadequacy in the manuscript. In our current study, time schedule following cerebral ischemia for each evaluation was only shown in the results and figure legends, but did not list in the methods. We had added the time schedule for each measurement in the methods according to your suggestion, and marked using yellow highlighting in our revised report (3.4. Sample Collection and Treatment).

  1. How about the ischemic penumbra sample, ischemic core sample, and sample far from the ischemia, but not 'brain sample' used in this study. The meaning of the obtained region is critically important.

Response: Thank you for the comment. Ischemic brain injury was located in the frontal lobe, including motor area and premotor area. The cerebral cortex was divided horizontally into six layers: the molecular, external granular, external pyramidal, internal granular, internal pyramidal and multiform layers [1]. Our study mainly focused in the cerebral cortex, and for each evaluation, the sample from cerebral cortex with tarnished or discoloured, which was considered as ischemic core, were collected to measurement, thus could ensure the accuracy of the results.

[1] Zheng, S.L.; Zhu, J.R.; Li, J.; Zhang, S.; Ma, Y.F. Leonurine protects ischemia-induced brain injury via modulating SOD, MDA and GABA levels. Frontiers of Agricultural Science and Engineering 2019, 6, 197-205, doi:10.15302/J-FASE-2018245.

  1. The number of rats used in this study should be described for each evaluation.

Response: We apologize for the inadequacy in the manuscript. We had added the total number of rats in “3.1. Animals and Focal Cerebral Ischemia Model”, and marked using yellow highlighting in our revised report. Moreover, the number of rats using in each group for evaluation were also listed in the figure legends (Figure 1-5), and marked using yellow highlighting in our revised report.

Reviewer 3 Report

The manuscript “Leonurine reduces oxidative stress and provides neuroprotection against ischemic injury via modulating oxidative and NO/NOS pathway” by Tang et al. explores the molecular mechanisms of alkaloid, Leuneurine in cerebral ischemia.

Major

According to the title the manuscript explores Leonurine. However, experiments were conducted with Leuneurine. In the manuscript, it is unclear whether both of them were used interchangeably or are different compounds.

Minor 

Was baseline activity in the open field tested?

For the behavior study, how was the randomization done? Was the observer blinded to the experimental groups?

Given 5 min as the total time to evaluate the open field activity, were mice given initial acclimatization time for the experiment?

Figure 1 – Given the nature of the experiment, the error bars are not visible in the graph.

Figures 2, 3, 7 – Y-axis should be rewritten. It is very challenging to interpret the results when the axis were not properly labeled.

Figure 4 – How the neuronal numbers were counted for each treatment group. How many sections per rat were taken for counting?

Please provide raw images for all the western blotting experiments.

Was there any effect of Leo itself on the levels of ROS/NO/iNOS?      

Author Response

Reviewer #3

Comments and Suggestions for Authors: The manuscript “Leonurine reduces oxidative stress and provides neuroprotection against ischemic injury via modulating oxidative and NO/NOS pathway” by Tang et al. explores the molecular mechanisms of alkaloid, Leuneurine in cerebral ischemia.

Response: Thank you for your support.

Major comments:

Major

  1. According to the title the manuscript explores Leonurine. However, experiments were conducted with Leuneurine. In the manuscript, it is unclear whether both of them were used interchangeably or are different compounds.

Response: Thanks to you for the comment. In the manuscript, we focused on Leonurine rather than Leuneurine. We checked up the manuscript carefully again and again, and Leuneurine did not occur in the report.

Minor

  1. Was baseline activity in the open field tested?

Response: Thank you for bringing up this issue. The system of open field tested using in this study mainly included “total distance”, “mean speed”, “time mobile”, “time immobile”, “max speed”, “peripheral region: time”, “peripheral region: distance”, “central region: time”, and “central region: distance”. In current study, we did not detect the baseline activity, such as the number of standing, and it will be carried out in the future research.

  1. For the behavior study, how was the randomization done? Was the observer blinded to the experimental groups?

Response: Thank you very much for your question. Firstly, the rats were randomly divided into four groups: CON group, sham group, Is group, and L-NAME group. Secondly, neurobehavioral test was determined by using the open field tested after the operation. The rats were transferred to the testing apparatus, which was an illuminated, soundproofed box (100 cm × 100 cm × 50 cm) with black inner walls. The bottom surface of the box was divided into 25 squares (20 cm × 20 cm). In order to ensure the accuracy of the results, each test rats from each group were measured as many as possible. Moreover, all of the rats from Is group and L-NAME group were detected. The observer was blinded to the experimental groups.

  1. Given 5 min as the total time to evaluate the open field activity, were mice given initial acclimatization time for the experiment?

Response: Thank you very much for your question. Before open field tested experiment, animals were adapted to the test room for more than 10 min. Then, the rats were transferred to the testing apparatus, and allowed 2 min to adjust. Finally, open field tested was conducted.

  1. Figure 1 – Given the nature of the experiment, the error bars are not visible in the graph.

Response: We are dreadfully sorry for the unclear pictures. According to your comments, we had been modified the Figure 1.

  1. Figures 2, 3, 7 – Y-axis should be rewritten. It is very challenging to interpret the results when the axis were not properly labeled.

Response: We appreciate your careful observation. According to your suggestion, we have changed the Y-axis in Figures 2, 3, 7.

  1. Figure 4 – How the neuronal numbers were counted for each treatment group. How many sections per rat were taken for counting?

Response: Thank you very much for your question. In current research, three slices were randomly selected for each group (five regions per slice) to count the neuronal numbers. We had added this information in the methods, and marked using yellow highlighting in our revised report (3.7. Histochemical and Immunohistochemical Procedure).

  1. Please provide raw images for all the western blotting experiments.

Response: Thanks to you for the comment. The raw images for all the western blotting experiments were provided in the later submission.

  1. Was there any effect of Leo itself on the levels of ROS/NO/iNOS?  

Response: Thank you very much for your question. As reported by our lab in last few years, Leo alone had no obvious impact on rats in the absence of cerebral ischemia, suggesting that it had low intrinsic toxicity. Additionally, treatment of Leo alone made no difference to the expressions of ROS/NO/NOS, to a level similar to that in controls.  

Round 2

Reviewer 3 Report

Please clarify – Based on groups mentioned in the methods section “Twenty-four rats were randomly divided into four groups: 1) CON: control; 2) sham: control treated with Rose Bengal, submitted to surgery without laser illumination; 3) Is: submitted to surgery for cerebral ischemia model; 4) L-NAME: treated with L-NAME, submitted to cerebral ischemia model”

Why Leo was not used for the in vivo experiments as the whole manuscript explores the role of Leonurine (Leo)?

As suggested previously regarding error bars, upon careful observation, no changes in Figure 1 were observed.

As suggested previously, no changes in the Y-axis were made in Figures 2,3 and 7

Please provide raw images of the western blot experiments

Author Response

  1. Please clarify – Based on groups mentioned in the methods section “Twenty-four rats were randomly divided into four groups: 1) CON: control; 2) sham: control treated with Rose Bengal, submitted to surgery without laser illumination; 3) Is: submitted to surgery for cerebral ischemia model; 4) L-NAME: treated with L-NAME, submitted to cerebral ischemia model”

Response: Thank you very much for your question. We apologize for the inadequacy in the manuscript. For “Histochemical and Immunohistochemical Procedure” study, twenty-four rats were randomly divided into four groups including CON group, sham group, Is group, L-NAME group, each with six rats. For “Oxidation and Antioxidant test”, “Measurement Of NO and NOS Content”, and “Western Blot Analysis”, twenty-four rats were randomly divided into four groups, each with six rats. However, for different time points studying of cerebral ischemia, including 2, 6, 12, and 24 hours, the rats were randomly divided into CON group, sham group, Is 2 h group, Is 6 h group, Is 12 h group, and Is 24 h group, each with six rats. Therefore, it is hardly to describe the total numbers of rats in the manuscript.

  1. Why Leo was not used for the in vivo experiments as the whole manuscript explores the role of Leonurine (Leo)?

Response: Thank you for bringing up this issue. Leonurine has recently been confirmed to be beneficial against ischemic stroke [1]. The salubrious effects of leonurine may be due various biological activities, such as anti-inflammatory, antiapoptotic and antioxidative activities [2, 3]. As reported by our lab in last few years, We found leonurine significantly improve the general activity of rats in an open-field test, which was associated with attenuated neuronal damage induced by ischemia. Moreover, serum SOD activity was significantly greater, MDA level lower in the leonurine group as compared with ischemia group [4]. But the data is too sparse to be sure the exact roles of leonurine, and therefore the specific mechanisms underlying its protective properties are still to be elucidated. To further refine our search, the aim of the present study was mainly to investigate whether leonurine could provide protection through nitric oxide (NO)/nitric oxide synthase (NOS)and oxidative stress pathway in in vitro experiments, combining with the pretreatment of the L-NAME. Apparently, it is worthy of further study on the anti-inflammatory, antiapoptotic and antioxidative mechanism of leonurine.

  • Loh, K.P.; Qi, J.; Tan, B.K.; Liu, X.H.; Wei, B.G.; Zhu, Y.Z. Leonurine protects middle cerebral artery occluded rats through antioxidant effect and regulation of mitochondrial function. Stroke2010, 41, 2661–2668, doi:10.1161/STROKEAHA.110.589895.
  • Liu, X.; Pan, L.; Wang, X.; Gong, Q.; Zhu, Y.Z. Leonurine protects against tumor necrosis factor-α-mediated inflammation in human umbilical vein endothelial cells. Atherosclerosis2012, 222, 34–42, doi:10.1016/j.atherosclerosis.2011.04.027.
  • Liu, X.H.; Pan, L.L.; Deng, H.Y.; Xiong, Q.H.; Wu, D.; Huang, G.Y.; Gong, Q.H.; Zhu, Y.Z. Leonurine (SCM-198) attenuates myocardial fibrotic response via inhibition of NADPH oxidase 4. Free Radical Biology & Medicine 2013, 54, 93–104, doi:10.1016/j.freeradbiomed.2012.10.555.
  • Zheng, S.L.; Zhu, J.R.; LI, J.; Zhang, S.; Ma, Y.F. Leonurine protects ischemia-induced brain injury via modulating SOD, MDA and GABA levels. Agr. Sci. Eng.2019, 6, 197–205, doi:10.15302/J-FASE-2018245.

  1. As suggested previously regarding error bars, upon careful observation, no changes in Figure 1 were observed.

Response: Thank you for your kindly reminder. According to your comments, we had changed the Figure 1, with clear error bars in the graph.

  1. As suggested previously, no changes in the Y-axis were made in Figures 2,3 and 7.

Response: Thanks for your patience to inform us of the inappropriate graph. We had removed the unclear description in Y-axis, and changed in a proper way.

  1. Please provide raw images of the western blot experiments.

Response: Thank you very much for your question. We conducted computer maintenance some time ago due to something was wrong. Unfortunately, some original files was not kept properly during copy process, including the original western blots for Bax and Bcl-2 of ischemia in rat. Additionally, only one original western blots for Bax and Bcl-2 of OGD-induced PC12 cells was copied in earlier. We are very sorry for that. If applicable, we could delete the results of Bax and Bcl-2 in the manuscript. Thank a lot, and once again my apologies for all the trouble.